# Water, Sanitation, and Hygiene Challenges in Informal Settlements in Kampala, Uganda: A Qualitative Study

**DOI:** 10.3390/ijerph20126181

**Published:** 2023-06-19

**Authors:** Julia Dickson-Gomez, Agnes Nyabigambo, Abigail Rudd, Julius Ssentongo, Arthur Kiconco, Roy William Mayega

**Affiliations:** 1Institute for Health and Equity, Medical College of Wisconsin, Milwaukee, WI 53226, USAakiconco@mcw.edu (A.K.); 2ResilientAfrica Network, Kampala P.O. Box 7072, Uganda; anyabigambo@ranlab.org (A.N.);

**Keywords:** WASH, informal settlements, diarrhea, slums, slum upgrading, water-borne infections, sanitation, solid waste removal

## Abstract

Diarrhea causes 1.6 million deaths annually, including 525,000 children. Further, chronic diarrhea puts children at risk for mineral deficiencies, malnutrition, and stunting which, in turn, can result in cognitive deficits, poor performance in school, and decreased disease immunity in adulthood. Most diarrhea is caused by water contaminated by fecal matter. Interventions to improve clean water and sanitation can save lives; however, challenges persist in informal settlements. In this study, we explored the views of residents of informal settlements regarding water and sanitation in their communities. Focus group interviews were conducted with residents of 6 informal settlements in Kampala, Uganda (*n* = 165 people), and 6 key informant interviews were conducted with governmental and nongovernmental organizations that work to improve informal settlements or provide services to them. The results from this study demonstrate that, although these informal settlements had many infrastructure “upgrades” such as latrines and toilets, water taps, wells, and garbage collection and drainage systems, the water, sanitation, and hygiene (WASH) system and its components largely failed due to point-of-use charges of water taps and toilets and the difficulty of emptying cesspits. Our results suggest that WASH must be considered a system and that multiple upgrading efforts are needed for WASH systems to work, including road construction and better oversight of fecal sludge disposal.

## 1. Introduction

Diarrhea causes 1.6 million deaths annually, including 525,000 children [1,2]. Further, chronic diarrhea puts children at risk for mineral deficiencies, malnutrition, and stunting which, in turn, can result in cognitive deficits, poor performance in school, and decreased disease immunity in adults [3]. Most diarrhea is caused by water contaminated by fecal material [4,5]. Interventions to improve clean water and sanitation can save lives. While this has been known for decades, many challenges persist, particularly in low- and middle-income countries (LMICs) and rural and informal settlements. In particular, while access to clean water has improved as a result of the Millennium Development Goals, over 2.5 billion people do not have access to basic sanitation, and over 1 billion people defecate in the open [6]. In Uganda, the proportion of people who practice open defecation is estimated to be 23% in rural and 9.4% in urban areas [7]; sanitation services are particularly lacking in informal settlements which house over 60% of the urban population [8]. Uganda has the highest rate of mortality for children under 5 years among all East African countries [9].

The population of LMICs is increasingly urban, placing burdens on housing and infrastructure resources. Currently, 56% of the world’s population—4.4 billion people—live in cities [10]. The urban population is expected to double its current size by 2050, at which point nearly 70% of the world’s population will live in cities [10]. Approximately 881 million of the world’s urban residents live in informal settlements, including an estimated 50–70% in Sub-Saharan Africa [11,12,13]. Informal settlements are often settled by people migrating from rural to urban areas in search of employment or to escape conflict [14], although many residents remain for decades [15]. Informal settlements or slums are defined by the United Nations Educational Scientific and Cultural Organization (UNESCO) as “a contiguous settlement where the inhabitants are characterized as having inadequate housing and basic services” [16] such as access to improved water, access to improved sanitation, sufficient living space, durability of housing, and security of tenure [14,17].

As a result of the unsafe and unhealthy living environments in informal settlements, there have been many efforts over the years to improve living conditions. These include improving or installing basic infrastructure such as water, sanitation, solid waste collection, electricity, storm water drainage, access roads, footpaths, and street lighting. Other improvements include improving housing or securing land tenure [18]. Less frequently, attempts to improve informal settlements have included developing educational and health services or public transportation in the settlements. However, little research has examined the effectiveness of these efforts in improving the health or socio-economic conditions of informal settlement residents. A recent Cochrane review [18] found five studies that evaluated the effects of efforts to improve informal settlements using a randomized controlled trial or a controlled before-and-after design [19,20,21,22,23]. They found an additional nine studies that used a control group but had only post-intervention data or that used an uncontrolled before-and-after design [24,25,26,27,28,29,30,31,32,33,34]. These interventions focused mainly on infrastructure improvements such as providing paved roads or sewerage systems. The studies, as a whole, found some improvements in diarrheal infections and other health outcomes, with mixed results in improving socio-economic status [18]. However, many of these projects are incremental, improving a single or a few sectors at a time [35].

Given the high burden of diarrheal diseases caused by contaminated water supplies [36,37], many projects have focused on improving water and sanitation [38]. However, the definition of “improved water” has many limitations, and so figures may exaggerate the actual extent to which people have access to clean water. For example, distance to communal water sources has been shown to affect how often people are willing to use taps or wells with a safer water supply [5]. Access to sanitation services, on the other hand, has not improved to the same extent as water [6]. Further, there is less evidence showing that improved water and sanitation infrastructure decreases diarrhea more than behavioral interventions such as handwashing and point-of-use water sanitation [4,5]. Part of the disappointing results may be related to the scale at which these services are provided, with some research suggesting that there may be too few water taps and toilets to accommodate the population and that they are too far away from households to be practical or are in disrepair [39,40,41]. Other studies have suggested that the price of emptying cesspits is too high for many residents, and so toilets may overflow [42], and that there are few accessible roads to toilets for sanitation trucks to empty them [42,43]. Further, research has suggested that improvements in water will have little impact without improvements in sanitation [4,5].

In this study, we explored the views of residents of informal settlements regarding water and sanitation in their communities. The results suggest that a more holistic approach must be planned for such improvements to be successful.

## 2. Methods

The research was conducted in 2 phases between May and June, 2021. In the first phase, we identified key stakeholders and conducted key informant interviews with them, focusing on the following topics: the nature of informal settlements in their countries and cities including their histories and formation; characteristics of the residents of informal settlements, including average age and proportion of children; health and other problems faced by residents; access to education for children in the settlements; different solutions proposed for informal settlements including past failed actions, current debates, and whether any proposed actions are currently being pursued; the best ways of providing services to existing informal settlements; and their opinions on barriers and the needed resources to address their cities’ and countries’ settlement crises. Key stakeholders included members from the Ministry of Settlements, Infrastructure, and Urban Planning; Ministry of Health; local and international nongovernmental organizations; and leaders/activists from informal settlements. All key informants were contacted by email followed by phone to schedule an appointment for a face-to-face interview. The email explained the purpose of the study, the nature of their participation, that their decision to participate or not was voluntary and would be kept confidential, and that their interview responses will be kept confidential. All provided written informed consent. A total of 5 key informant interviews were conducted.

In the second phase of the project, we held focus group interviews with members of informal settlements from Kampala. We purposively sampled residents to match the demographic characteristics of adult residents of the settlements and held separate focus groups for men and women, youth, and refugees. In the focus groups, we asked residents to describe how the settlement was formed and how they came to live there, their perceptions of the biggest problems facing the community and their children, their perceptions of the responsiveness of government and nongovernmental organizations in trying to address the problems or to force them to leave, and the best ways of providing health services to their community. Inclusion criteria: a person was included in the study if they were 18 years or older, living in the informal settlement, and able and willing to provide informed consent.

A total of 173 participants were contacted and interviewed. Of these, 8 were key informants, 165 participated in focus group discussions (83 women, 82 men), and 3 participated in in-depth interviews, as indicated in Table 1 below. Focus groups were held in six informal settlements, namely, Bwaise, Kifufumbira in Kawempe division, Namuwongo in Makindye division, Kinawataka and Naguru in Nakawa division, and Kisenyi in the Central division.

### 2.1. Analysis

All interviews were transcribed verbatim and translated into English when necessary. The analysis started by the reading and rereading of the manuscript by 2–3 people who developed as many codes as possible (open coding). Then, the codes were cleaned and collapsed. Then, the lists from different reviewers discussed to develop the final code sheet. Then, experienced research assistants attached codes to each meaning unit in all the interviews. Items with similar meanings were collapsed into sub-themes, and then sub-themes with similar meaning were collapsed into themes. Findings were summarized using narratives, supported by verbatim quotations from the data.

#### 2.1.1. The Credibility of the Study

The focus group discussion (FGD) guide and the in-depth interview (IDI) guide were translated from English to Luganda, and the quality of the translation was assured through back-translation. Subsequently, the tools were pretested to eliminate ambiguities in the guides before the actual data collection. The research assistants were given additional study-specific training, encapsulating the study aim, objectives, procedures, and ethical issues. Each interview took about 45 min to 1 h to be completed. The presentation of the findings is through verbatim quotations.

#### 2.1.2. Validity

The information gathered was related to the challenges of accessing and utilizing WASH services within the informal settlements. The participants expressed their views on their experiences of WASH challenges in the informal settlements. At the inception, interview rapport was established, and all participants consented with the autonomy to decide either to participate, freely express their views, or decline participation.

## 3. Results

### 3.1. Water

#### 3.1.1. Improved Water Was Available but Costly

Participants from all communities reported access to “improved” water. Most often, improved water was provided through public taps that could be accessed and paid for with electronic “keys”. The cost of access was around UGX 100–200 or approximately USD 0.01–0.02. While very low, some participants reported that this cost discouraged many to use taps.


*I just want to enlighten my friend [number 3]… If you see free things next to you and another that is not, which one will you go with? For example, if you sloped down there to the chairman’s place…. But someone will say no instead of spending my shs100 supposing I buy 6 jerry cans. I will be spending money for food. They do not want to waste money because now this kind of situation that we are living in is to always be very cautious in spending.*

*(Men’s focus group, Naguru.)*


The economic situation was made more difficult by the COVID-19 pandemic as many residents were left without work due to government curfews. Added to this, residents also explained that costs for the taps were raised without explanation during the COVID-19 pandemic.


*We would be accessing clean water, but all taps are expensive, the water costs were raised. To get a jerrycan of water now costs 200 shillings. Remember with corona virus people no longer have money and some taps charge 300 shillings. Even when you go to the well, you find so many people and there are also people who have taken charge of the well and when you go there, they want to sell the water to you at 500 or 1000 shillings a jerrycan. We do not really have clean water because the costs were raised because of the government charges we hear but we are really not sure of the reasons why!*

*(Women’s focus group, Kifumbiera, Kawempe.)*


Some residents had tap water accessible from their homes, but this was often unaffordable to many residents.


*In terms of water, there are several water sources. We have water where we use sensor keys, but it is not accessible to everyone because they are few and tap water here is very expensive. Me myself here, we had tap water, but it was very expensive to sustain so we had to cut the water off leaving people to move to far distance to find water… In terms of hygiene, it has not been good at all because people are negligent everyone minds their own business hoping for KCCA to come and clean for them leaving the water to only a few voluntary people who do not also sometimes have time to do that work.*

*(Youth focus group, Naguru.)*


As mentioned by the participant above, water taps were often located far from people’s homes. In addition, water supplies from taps are not reliable.


*On the issue of water, much as we have national water access, it is on and off like right now there is scarcity. As we talk now people are roaming with jerry cans. We don’t know if the roads that are being fixed are causing the supply pipes to be broken.*

*(Men’s focus groups, Kifumbira.)*


Participants also talked of buying jerrycans to purchase water from people with taps at UGX 100 to 200 a can.

Other water sources people reported using included wells, although people recognized that water from these sources was not safe to drink without boiling.


*When national water is off, we rely on underground wells which are very contaminated due to the nature of our environments. Sometimes someone can pour urine into the open well. Previously there was a research that was done on the water we use in Kampala and most of the wells were found extremely contaminated. Due to high population, the water table must be contaminated because even when you just taste or smell water from wells here, the water tastes different.*

*(Men’s focus group, Kifumbira.)*


#### 3.1.2. Home-Based Practices to Improve Water for Drinking Were Not Widespread

Although participants were clear that boiling water could reduce waterborne illnesses, it was necessary for residents to buy cooking fuel (most often wood or charcoal) in order to boil it, incurring an additional expense. Many participants reported only boiling water they suspected was unclean.


*The challenges we get whenever the water is dirty but without rain, the water is clean. So, when we get that water, we boil water for drinking.*

*(Women, Namuwongo.)*


No participants mentioned using chemical water treatment, such as chlorine. Because of the problems with wells being contaminated, some borehole wells were locked and not accessible to people at the times they needed it.


*The water point is closed since 7 am and opened at 1 pm. Imagine it is you and the borehole has been closed because that borehole came up as a project, it is a borehole, but they close it in the morning hours and opened when it is time for prayers now where will you fetch the water from? Will not you go down and fetch that near the drainage. So that is the point, get some time pass by and see.*


### 3.2. Toilets

#### 3.2.1. Pit Latrines Are the Dominant Type of Toilet and Are Overcrowded

Most participants attributed problems with water cleanliness to problems associated with toilets and other sanitation problems.


*There used to be just large tree above the well, up until the palace. When we come to the science aspect of it, the roots of a tree in a way cleanse water, and also, there was no single latrine between the well up until the palace. But over time, the latrines were constructed on the slopes and the fecal matter goes down to the soil and yet there are no trees on the slopes anymore. So, when the water also flows, it mixes with this fecal matter together with the urine already in the ground down to the well. With that, the taste of the water changes, together with its coolness. In the past, you would go to the well to fetch water in the afternoon and you would find it so cool like it was from the fridge unlike now- that is the bad the new residents came with.*

*(Women’s focus group, Kisenyi.)*


The toilets available are largely pit latrines and include both private and public latrines. Private latrines were built by owners of buildings who sometimes, but not always, allowed their tenants to use them. As overcrowding is common, with multiple families occupying just a few rooms, latrines often became overburdened and unclean.


*Toilets are available but they also not clean because in the ghetto you will find that one room has six people or four. A man, mother, and the children, yet you the landowner will not tell them that I need three people in my house because you also need the money. You can wake up early and clean the toilet, then they spoil it, and you will not be always arguing with each other. You just look over it and keep quiet. Otherwise, sanitation is very poor.*

*(Men’s focus group, Naguru.)*


In addition, not all houses included latrines, and those that are built are not always well placed.


*The main problem is the lack of toilets in my place. People construct houses so close to each other and there are no toilets. You can find 5 houses with no toilets. KCCA constructs a toilet in one area, for example, for Nakawa division, you may find a toilet in Luzira, implying people from Kinawataka or Mbuya 1 miss out. So, our humble cry is about the toilet. We need toilets. We have only two toilets which are far at the extremes. There is no toilet in the center.*

*(Women’s focus group, Kinawataka.)*


#### 3.2.2. Cesspits Must Be Emptied and Toilets Cleaned, Introducing Costs to Informal Settlement Residents

As the toilets are pit latrines with many people using them, cesspits fill frequently. These must be emptied, most often by private companies. In order to recoup these costs, many landlords charge for the use of the toilets. However, many toilets are inaccessible to sanitation trucks as the roads are impassable.


*On that issue, I suggest that if someone is in a slum, they can’t allow them to construct toilets for each person because, everyone has a small plot. If you are to construct toilets, you find rooms so close and you may find toilets are more than houses. And even then, there are no roads, if the toilets get blocked, there are no roads for the vehicle to go and open them up. And even those people that work in the public toilets, I support them because if you allow all children to go to the toilet, it gets full in one week. We had a “Toilet for Hope”, which was for the children, it gets full in one week. Emptying it was shs.200, 000 and the responsible person needs money to empty the toilet and pay the workers and water, so that money couldn’t be raised. That is why they chase the children and suggest that at least children pay shs. 100 and adults shs.200, so that life goes on, but also what would work best would be, for low-income earners to get shs. 3000 for a month. Because the daily payments may not be available.*

*(Women’s focus group, Yoka, Namuwongo.)*


Paying for each use means that when people do not have money, they may have to resort to open defecation or using plastic bags. Informal settlement residents are often employed in the informal sector, and so daily income is often variable and uncertain.

Participants also reported that in some areas where toilets are inaccessible to sanitation trucks, landlords may empty them directly into the drainage canals, eliminating any sanitation benefit of having the toilets in the first place.


*Interviewer: Does that mean that they are functioning well or no?*



*No, they are not working well but for people to avoid emptying their toilets because trucks cannot reach down there, they just release it into drainage channels. There is also a time when the owner here released the sewage in the channels and we the LC were made to clean the channel because after doing it, went to Mukono then the police and KCCA came and made us to clean the drainage channel.*

*(Women, Naguru Go Down.)*


In addition, as many participants pointed out, the water table in the informal settlements was very high, and toilets were often destroyed by water at the same time as they contaminated the water supply.


*About toilets, they would have been constructed but given that we are in a swampy area, the moment you dig about two feet underground, you find water. Even those that have constructed the toilets and cemented them face a problem of water given that the water table in our area is high, the water penetrated through the cement.*

*(Youth, Namuwongo Makindye.)*


Public toilets often rely on community residents to keep them clean. As many participants explained, the toilets often became filthy to the extent that people did not want to use them much less clean them.


*I also want to talk about the toilet problem. I want to support what she said but also the people with toilets which would have been clean and well, given that we have people from various backgrounds Kisoro, Arua, Buganda and so on, so people just do not care about cleaning the toilets… There are also houses that do not have toilets but because when you come to the community and rent a room at 50,000 shillings, they tell you to find means of disposing of human waste and in the end, you adopt using a bucket during the day then in the evening you pour everything in the drainage channel. That is why people have been talking about the water at the well! It is because of this drainage channel. Sometimes, there are people also who open their toilets and release sewage through the drainage channels which affects water. But about cleaning toilets, we are a mixed culture, so every tribe comes with their own practices. There are individuals that tell you that their culture does not permit them to dump human waste in the toilet holes so, when they enter them, they dump on the sides and if someone else comes and finds that mess, they also dump on the other side. So, people are unruly, and people do not care for their lives be it male or female.*


#### 3.2.3. Toilet Problems Lead to Pollution of Drainage Canals

In cases in which toilets were too dirty to tolerate, participants reported using other means such as buckets or polythene bags which they would then throw onto roofs or dump in the drainage canals.


*Interviewer: So, for those that are full, where do the community members go to ease themselves?*



*People then resort to using polythene bags and then dump it anyhow because they do not have where to go. One day I visited a friend within our community and nature called, I had to use the nearest toilet that was near but when I got inside, for sure it was full with maggots but people were still using it to an extent that they sought for the slightest space to place their waste. It is actually a raised toilet but also cracked because I would look through the cracks. When I got there, I just lost the appetite and had to go elsewhere.*


Others resorted to polythene bags because the per-use cost charged by landlords to use private toilets and, occasionally, public toilets was too expensive, particularly for families with many children.


*We have challenges with toilets. In one way or another, majority people do not use toilets because mostly toilets are at a cost yet most of the children cannot get that money say 100 shillings to go to the toilet. What most people do is, they defecate in the polythene bags, and they throw them in open areas. You may find some places stuffed with feces. That is the challenge of poverty because people cannot afford to pay for toilets.*

*(Youth, Namuwongo Makindye.)*


Some settlements have rules that allow children and people with diarrheal infections to use the toilets for free. However, many times people are still charged to use the toilets, resulting in people defecating in the open or using polythene bags precisely when sanitation is most needed to prevent the spread of waterborne illnesses.


*Adding on the points of [other respondent] cleanliness in this area has two points: we are from various walks of life, people here might make up to 11 tribes, you find someone who can say going to a toilet I would rather buy my polythene bags of 1500 shillings and use them the entire moth instead of paying for the toilet. The local council put a law that if they get you, you pay a fine or clean the drainage. We have people who have their issues like diarrhea, if someone goes to the toilet more than twice in a short time then it becomes free. Also, we don’t charge those who come with water and toilet paper. Young children do not pay, we basically charge those who do not come with their own water and toilet paper.*

*(Youth, Katanga.)*



*Ha, some people are dishonest! One time I had diarrhea and I had to pay 2000 shillings, the person I found told me to pay at once since he was tired of collecting money from me every after a few minutes.*

*(Youth, Katanga.)*


### 3.3. Garbage

#### 3.3.1. Collection Points for Garbage Were Few and Often Inaccessible

The collection of solid waste, while it exists, has many system failures that cause trash to accumulate and eventually end up in drainage canals. Participants complained that there were few collection points for garbage which meant that many dumped garbage in the canals. This was particularly the case for people who lived away from main roads where trucks cannot pass.


*We shall go back to our community (Naguru), still communication, you find those who bring garbage are the ones close to the road but the rest just dispose of into the drainage channels, yet the very channels connect to lake Victoria. You will find that people from here put it in the channels and wait for the rain but those that are near here they take it to the collection site…. If there are no roads in a slum of this kind, no work done really. Let us leave it as that because what we put in up here in terms garbage collection, it is very little. Without roads garbage, it is very hard carrying garbage from here up to the tarmac even if you go to Kasokoso our neighbors, it is a big slum like ours.*

*(Men, Naguru.)*


#### 3.3.2. Current Financing Models for Garbage Collection Were Not Sustainable

In some communities, residents paid private companies to collect trash. However, if one household does not pay for garbage collection, often these companies discontinued garbage collection for the whole neighborhood.


*Sometimes there are groups where members subscribe or even people can decide on a day to clean the community, but we currently have women groups that sometimes come together and agree to go and do general cleaning in communities… but the challenge is that KCCA is the main cause of this problem because you cannot get a car from them. Their cars are never at station, when you go to them, they always say, “The vehicles do not have fuel, they are not available”. There is an organization that was willing to provide the service by providing us with a car to collect garbage from the community, but KCCA frustrated them and told them that, “If you bring a truck, you have to give it to us” and the organization also refused and said, “If we bring a truck, it belongs to us!” So, they disagreed and the organization pulled out but the whole problem is with KCCA. You can collect garbage and it takes a long time without the cars coming to pick it until it even starts to get back to peoples’ homesteads. That is the challenge that we are facing.*

*(Women, Kifumbira.)*



*Interviewer: [Number 8], you mentioned that you are part of the area leadership, is there a way that the local council can deal with the garbage here?*



*We failed to find a way due to the following reasons, they started charging a fee so people can’t afford, they decide to dump into the drainage channel. In my opinion the division is the one responsible for such a big role. You see even the landlords have few toilets, when people dirty the toilet they do not want to clean so they stop using the toilet. Then they resort to kaveera (polythene bag). We have really tried on the garbage issue but we have failed. That three thousand shillings is too much.*

*(Women, Bwaise.)*


As the participant above comments, polythene bags full of feces due to the lack of toilets are part of the garbage that blocks the drainage canals. While the KCCA does have garbage collection points in informal settlements, they are often located far from some residents, and pick-up times are infrequent and unpredictable.

#### 3.3.3. Poor Garbage Collection Exacerbates Flooding: Garbage and Fecal Materials End up in Homes

As garbage and trash are often dumped into drainage canals, they are often blocked. During the rainy season, this means that many of the homes become flooded, and feces and garbage that are dumped into the canals end up in people’s homes.


*There are drainage channels for water, but because people are reckless, sometimes they pour garbage there. So, the channels are messed and when it rains what others do is to release fecal matter into the water channel. Sometimes you find the rains are light and do not drain off the fecal matter and it stinks for people. Like [other participant] said that some time people have garbage at their door, in most cases they don’t take the garbage to the right place. Even the people that are paid shs.1000 because it is what they want, but just empty the garbage into the drainage channel. So, they remove garbage from here and through it into the water channel and they don’t care.*

*(Women, Yoka.)*


## 4. Discussion

The results from this study demonstrate that, although these informal settlements had many infrastructure “upgrades” such as latrines and toilets, water taps, wells, and garbage collection and drainage systems, the WASH system and its components largely failed. Participants in our study were very clear about the interconnection of different components of WASH and other infrastructure such as roads and recognized that the failure of some components, such as clean latrines or trash collection, led to the failure of other systems such as drainage. Some of these failures may have come from piecemeal approaches, for example by constructing toilets without consideration for roads that are needed to empty cesspits. These results help explain why some meta-analyses have found that improved water has no effect on diarrhea disease unless it is accompanied by improvements in sanitation [44].

Some research has suggested that upgrading slum settlements is preferable to constructing new affordable housing developments [35,45]. However, the results from this research suggest that there is a lack of space in informal settlements to upgrade in a way that will avoid these failure points in the WASH system. Respondents reported that settlements were already crowded with latrines that could not meet the demand of the number of people in the settlements. They also complained that water tables are high, and cesspits must be emptied frequently; however, cesspit emptying is best performed mechanically with trucks which cannot access the latrines due to the lack of roads. Constructing roads, on the other hand, would require the displacement of some residences as existing paths are widened. In other words, some amount of displacement seems inevitable for the system to work as it is.

As pit latrines cause a number of problems in the settlements, other forms of toilet should be explored. Many researchers have argued that flush toilets connected to a sewerage system is economically out of reach for most residents of slum settlements and for LMIC governments [6]. Wittington and colleagues estimate that the costs of operating flush toilets to be over USD 50 to 100 per person, significantly more than most families can afford [46]. Further, such systems require water treatment plants which are expensive to construct and to maintain. Finally, such systems use enormous amounts of water, also in short supply in informal settlements and in LMICs. They suggest, as solutions, that sanitation services empty cesspits as a business; however, as they also point out, many pits must be emptied by hand, exposing workers to health risks, and many times fecal sludge, whether emptied manually or mechanically, is not emptied properly and ends up in landfills or drainage canals [6]. Indeed, research on the sanitation markets in East Africa has suggested that services to empty cesspits are too expensive, and the poor maintenance of trucks, poor quality vacuum pumps and pipes, and poor maintenance of pit latrines are considerable bottlenecks to sludge removal [42,47,48], although Kenya found a social enterprise, Sanergy, to be effective in emptying pits [49].

As in other research, participants in this study mentioned that the cost for using toilets was often too high, particularly as it was charged per use. Participants in our study weighed the costs and benefits when asked to pay for each use of water tap or toilet. Often for these participants, it was a choice between paying for water from the tap or to use the toilet or paying for food. Residents’ precarious economic situation may have been exacerbated by the COVID-19 situation, but environmental disasters or other shocks occur frequently in some LMICs, and informal settlement residents are the most vulnerable to shocks. In extreme vulnerability, and with other free if not ideal ways to obtain water or defecate, people will most often choose to hold on to their money to pay for food. The likelihood of choosing not to pay for toilet or tap use seems to be exacerbated by point-of-use charging. In the urgency of the need to use the toilet, people short of money will be forced to defecate outside the toilet.

Another solution is the replacement of pit latrines with composting toilets that would use biological processes to decompose human waste that could then be safely used for other purposes such as fertilizer or fuel. There are several such toilets along the spectrum of proof of concept to prototype. Such systems would be water-saving and thus may be more environmentally sustainable in our changing climate. There is an urgent need to pilot these reinvented new toilets in low-resource settings [6].

Some research has suggested that behavioral interventions may be more effective than changes in infrastructure to reduce diarrheal infections [4,5,6,50]. These include point-of-use water treatment and handwashing which have been found to substantially decrease diarrhea incidence [4,5]. Many social and behavioral change strategies for WASH improvements have been successfully implemented. For example, community-led total sanitation (CLTS) uses collective behavioral change to create “open defecation free” communities [51]. However, our results suggest that open defecation is not always a choice. Further, the extent to which such behavioral change can be sustained is unclear, and improvements often dissipate over time. Residents reported that various organizations had come to teach people about hygiene, and people generally seemed to be aware that boiling water would make it safe to drink; however, they did not always take the time or use the fuel to do so as boiling water also incurred a cost. No participants mentioned chemically treating water or receiving chlorine or other chemicals to do so. Receiving free or highly subsidized chlorine has been attempted in some communities [4]. Further, some residents considered some of their water sources “safe”. Some research has suggested that providing dispensers at the point of water collection could overcome the low uptake of such methods [4,50]. However, this may allow water to be contaminated en route to the point of use. Although such approaches may decrease diarrheal infections, residents themselves prioritized fixing their toilet system.

Contributions in the form of work to keep wells or public toilets maintained often failed according to participants. Research has found that maintenance of rural water infrastructure has been a continuing problem, with up to 50% of systems in disrepair [52]. Little research has tested different ways to manage water infrastructure such as community organizations; local, municipal, or federal management; or private management, and what little research has been conducted has often not compared different methods, making conclusions difficult [4,5]. To our knowledge, no research has tested different management systems for public toilets or communal water supplies in informal settlements. However, our results suggest that various methods are used including private owner control, private emptying services, the KCCA, and community volunteers. People complained that those assigned to maintaining these services did not do so. In one case, after failing to raise money for emptying the cesspit, those in charge of managing a public toilet resorted to charging per use which brought its own difficulties. The problems of free riding were also evident for people who paid for trash collection but received no services as others were not willing or able to pay for the services.

## 5. Future Research

Future research to compare different ways of managing water and sanitation in informal settlements is needed. In particular, research regarding what is affordable and what residents are willing to pay for water and sanitation is needed. Willingness to pay studies may not give an accurate picture of what is affordable to residents; water and sanitation are seen as necessities to informal settlement residents, and they may over-estimate what they are able to pay without sacrificing other necessities such as food, shelter, and health care. Rather, research should measure people’s actual use of water and sanitation facilities and pilot tests of new technologies for sanitation, such as waterless toilets to replace pit latrines in real-life settings.

## 6. Limitations

The results of this research are limited to residents living in the informal settlements included in this study and may not be generalizable to residents of other informal settlements in Kampala or in other countries.

## 7. Conclusions

The results from this study suggest that WASH upgrading has been inadequate to meet the demands of informal settlement residents. Further, point-of-use charges for toilets and water taps made them economically out of reach for many residents. Water and sanitation services need to be more heavily subsidized to provide informal settlement residents with access. Our results suggest that WASH must be considered a system and that multiple upgrading efforts are needed for WASH systems to work, including road construction and better oversight of fecal sludge disposal. Alternatives to latrines or flush toilets are urgently needed to address the sanitation problems in informal settlements.

## Figures and Tables

**Table 1 ijerph-20-06181-t001:** Stratification of study participants by gender and study site for informal settlements project.

Type of Interview	Gender	Government/NGO	Kawempe	Makindye	Nakawa	Central	Total
Key Informant Interviews	Male	4	2	0	2	0	8
	Female	0	0	0	0		0
	Total	4	0	0	1	0	5
Focus Group Discussions	Male	0	28	17	20	17	82
	Female	0	26	9	28	20	83
	Total	0	54	26	48	37	165
	Total	0	2	0	1	0	3
Overall total		4	56	26	50	37	173

## Data Availability

Data are available from the first author by request.

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
