# Peer review of "Water, Sanitation, and Hygiene Challenges in Informal Settlements in Kampala, Uganda: A Qualitative Study"

_ijerph, 2023, doi:10.3390/ijerph20126181_

Round 1
Reviewer 1 Report
In the introduction is important to describe what is WASH system and the implications for water quality.
1. What is the main question addressed by the research? The main idea of this research is to analyze the actions of the wash system in informal settlements.
2. Do you consider the topic original or relevant in the field? Does it
address a specific gap in the field?
The topic is relevant but not original. They looked forward solutions but the metodology used was week to answer the big question.
3. What does it add to the subject area compared with other published material?
The analyzes carried out reiterate the need to seek the quality of the water used in settlements
4. What specific improvements should the authors consider regarding the
methodology? What further controls should be considered?
A control group could have been used for comparison. For example, a settlement with some established and tested water treatment method.
5. Are the conclusions consistent with the evidence and arguments presented
and do they address the main question posed?
The arguments of the discussions are pertinent. However, I did not observe that the methodology used was able to explain the question pointed out.
6. Are the references appropriate? Yes
7. Please include any additional comments on the tables and figures.
I have no comments to make about the table
Author Response
“The topic is relevant but not original. They looked forward solutions but the metodology used was weak to answer the big question. The analyzes carried out reiterate the need to seek the quality of the water used in settlements.”
While we agree that our results are very consistent with other studies, we respectfully disagree that the methodology was weak. We used a very thorough sampling plan and conducted several focus group and in-depth interviews.
“A control group could have been used for comparison. For example, a settlement with some established and tested water treatment method.”
We conducted focus group interviews in 6 different informal settlements and have presented the results from all of them. Our results show that there is not, in fact, a settlement with an established and tested water treatment method.
Reviewer 2 Report
This is an interesting paper focusing on WASH challenges in informal settlements in Kampala, Uganda. I have few suggestions/comments to make with sole aim of improving the final version of this manuscript.
From the title after going through the entire manuscript, I found it will be necessary to include "aqualitative survey approach".
From the abstract, give full meaning of WASH at first mention. Why did you think WASH systems ant it's components failed since you have mentioned the fact that the have witnessed infrastructure upgrades?
You may need to reconstruct your recommendation now.
From the introduction, give full meaning of LMICs at first mention. There is a need to present a more robust background information for this study especially in adding information regarding;
(I) rank of Uganda in open defecation problem
(Ii) how many children are at risk of chronic diarrhea in Uganda?
(iii) what's the state of WASH in Uganda in recent time (2022-2023). These are germane information that need to be presented in this section.
In line 88-89, kindly move this to the results discussion section.
In the methodology, kindly add the month and year of this survey. In your FGDs, add the figure for men too.
The conclusion and recommendations section is conspicuously missing in this manuscript. Kindly include it. Do not merge the limitations of the study with results discussion. You may need to separate it. Further, add "areas for further studies".
Thank you.
Author Response
Reviewer 2. This is an interesting paper focusing on WASH challenges in informal settlements in Kampala, Uganda. I have few suggestions/comments to make with sole aim of improving the final version of this manuscript.
From the title after going through the entire manuscript, I found it will be necessary to include "aqualitative survey approach".
We have added “a qualitative study” after the semi-colon.
From the abstract, give full meaning of WASH at first mention. Why did you think WASH systems ant it's components failed since you have mentioned the fact that the have witnessed infrastructure upgrades?
We have spelled out WASH at the first mention in the abstract and body of the text. We have also included reasons the WASH system failed.
From the introduction, give full meaning of LMICs at first mention. There is a need to present a more robust background information for this study especially in adding information regarding;
(I) rank of Uganda in open defecation problem
(Ii) how many children are at risk of chronic diarrhea in Uganda?
(iii) what's the state of WASH in Uganda in recent time (2022-2023). These are germane information that need to be presented in this section.
We have spelled out LMICs at first use and added information regarding open defecation and risk of diarrhea in Uganda. We were not able to find more recent information on WASH (2022-2023).
In line 88-89, kindly move this to the results discussion section.
We believe that this literature belongs in the introduction as it covers problems found with pit latrines in informal settlements in a number of contexts.
In line 88-89, kindly move this to the results discussion section.
We believe that this literature belongs in the introduction as it covers problems found with pit latrines in informal settlements in a number of contexts.
In the methodology, kindly add the month and year of this survey. In your FGDs, add the figure for men too.
We have added these.
The conclusion and recommendations section is conspicuously missing in this manuscript. Kindly include it. Do not merge the limitations of the study with results discussion. You may need to separate it. Further, add "areas for further studies".
We have added a conclusion, limitations and future studies sections.
Reviewer 3 Report
The article has an original and important approach on a subject that is of the greatest importance today.
Note, however, the following:
All abbreviations and acronyms must be explained (in full) the first time they appear in the text (eg WASH, LMIC, first column of Table 1, etc.).
Is the formatting of lines 127 and 128 correct?
A Conclusions chapter is missing (which may incorporate some final paragraphs from the Discussion chapter).
The bibliography seems adequate.
In order to help the authors improve their manuscript, and inform the Academic Editor's decision, please consider providing some additional, specific comments such as:
1. What is the main question addressed by the research?
This article seeks to analyze why many initiatives to promote safe water supply, sanitation and hygiene conditions fail, especially in the most problematic regions. The research method is based on interviews, which allows for a different view from what is generally presented, based on theoretical analyzes that sometimes prove to be out of step with reality.
2. Do you consider the topic original or relevant in the field? Does it address a specific gap in the field?
Although the topic is not original, the approach is very interesting, relevant and novel.
3. What does it add to the subject area compared with other published material?
Indeed, this work draws attention to the difference that exists between theoretical concepts and the results of their practical application, namely with regard, in this case, to solutions to guarantee the safe supply of drinking water and adequate sanitation. The United Nations SDG6 objective, perhaps the most important one, does not currently show the necessary progress and in this article some clues can be found that can justify some situations.
4. What specific improvements should the authors consider regarding the methodology? What further controls should be considered?
I consider that the methodology is adequate, although it has potential for improvement.
5. Are the conclusions consistent with the evidence and arguments presented and do they address the main question posed?
A Conclusions chapter is missing in the paper, although in the Discussion chapter some paragraphs already contain some conclusions.
6. Are the references appropriate?
Yes
7. Please include any additional comments on the tables and figures.
The use of abbreviations and acronyms in Table 1 must be preceded by their previous explanation in full in the text.
Author Response
Reviewer 3. The article has an original and important approach on a subject that is of the greatest importance today.
Note, however, the following:
All abbreviations and acronyms must be explained (in full) the first time they appear in the text (eg WASH, LMIC, first column of Table 1, etc.).
We apologize for this oversight and now spell out the acronyms the first time they are used.
Is the formatting of lines 127 and 128 correct?
If this refers to the table, it seems to not fit in the margins of the paper but I can’t seem to reformat it on the template provided.
A Conclusions chapter is missing (which may incorporate some final paragraphs from the Discussion chapter).
We now include a conclusion.
Round 2
Reviewer 2 Report
The editor can make final publication decision on the revised version of this manuscript. Thank you.